# A Continuum Damage Model for Intralaminar Progressive Failure Analysis of CFRP Laminates Based on the Modified Puck’s Theory

**DOI:** 10.3390/ma12203292

**Published:** 2019-10-10

**Authors:** Jiefei Gu, Ke Li, Lei Su

**Affiliations:** 1Jiangsu Key Laboratory of Advanced Food Manufacturing Equipment & Technology, Jiangnan University, Wuxi 214122, China; 2State Key Laboratory of Mechanics and Control of Mechanical Structures, Nanjing University of Aeronautics and Astronautics, Nanjing 210016, China; 3School of Mechanical Engineering, Jiangnan University, Wuxi 214122, China; lei_su2015@jiangnan.edu.cn

**Keywords:** CFRP laminate, continuum damage model, intralaminar progressive failure analysis, modified Puck’s theory, in situ strength effect

## Abstract

A continuum damage model is proposed to predict the intralaminar progressive failure of CFRP laminates based on the modified Puck’s theory. Puck’s failure criteria, with consideration of the in situ strength effect, are employed to evaluate the onset of intralaminar failure including fiber fracture and inter-fiber fracture. After damage initiation, a bilinear constitutive relation is used to describe the damage evolution process. In strict accordance with Puck’s concept of action plane, the extent of damage is quantified by the damage variables defined in the fracture plane coordinate system, rather than the traditional material principal coordinate system. Theoretical and experimental evaluation of CFRP laminates under different loading conditions demonstrates the rationality and effectiveness of the proposed numerical model. The model has been successfully implemented in a finite element (FE) software to simulate the intralaminar progressive failure process of CFRP laminates. A good agreement between the experimental and numerical results demonstrates that the present model is capable of predicting the intralaminar failure of CFRP laminates.

## 1. Introduction

Carbon fiber reinforced polymers (CFRPs) are being increasingly used in industry due to their advantageous properties such as high specific strength and stiffness, good resistance to fatigue and corrosion, as well as flexibility in design. CFRP laminates are now widely applied in various fields, including aerospace structures, windmill blades, and pressure vessels. Nevertheless, the design of CFRP composite structures is still rather conservative in engineering practice. A large quantity of time-consuming and expensive tests must be carried out to ensure structure safety. To efficiently reduce the time and cost as well as fully exploit the advantages of CFRPs, there is an imperative need for the accurate theoretical prediction of the failure of CFRP laminates. However, the failure mechanisms of CFRP laminates are very complex due to their inherent anisotropy and the variety of failure modes. Hence, developing a reliable failure theory for CFRP laminates is a very challenging task.

The failure process of CFRP laminates can be divided into two stages. First the damage initiates in a ply, and then it gradually propagates through the laminate until the structure reaches the ultimate failure load [1]. The continuum damage mechanics (CDM) approach, which was originally developed by Kachanov [2], has been successfully employed in the progressive failure analysis of CFRP laminates and achieved good results [3,4,5,6,7]. The CDM approach uses failure criteria to predict damage initiation. Once damage initiates, the material stiffness will be degraded to simulate damage propagation. The reduction of the stiffness coefficients is controlled by the damage variables, which increase monotonically as damage accumulates.

A large number of failure criteria have been developed to predict the damage onset of CFRP laminates. Classical criteria such as the maximum stress or strain, Hoffman, and Tsai–Wu fail to distinguish the intralaminar failure modes, namely the matrix failure and the fiber failure. In 1980, Hashin proposed a set of criteria which separate the fiber and matrix failure modes [8]. Although Hashin’s criteria have been widely used in research [9,10,11], they are incapable of accurately predicting the matrix damage in compression [12]. Nevertheless, Hashin’s assumption that material failure is exclusively caused by the stresses acting on the fracture plane is physically meaningful. Inspired by Hashin’s insightful idea, Puck and his co-workers developed a new failure theory for composite materials [13]. Puck’s failure criteria distinguish two different types of fracture: inter-fiber fracture (IFF) and fiber fracture (FF). Inter-fiber fracture comprises both cohesive matrix fracture and adhesive fracture of the fiber/matrix interface [13] (p. 7). In this mode a macroscopic crack runs parallel to the fibers through the thickness of a layer; see Figure 1. In the fiber fracture mode, composites fail due to fiber rupture. In the first world-wide failure exercise (WWFE-I), Puck’s failure theory was among the five leading theories recommended by the organizers [14]. In the second world-wide failure exercise (WWFE-II), it was judged to be a fairly developed model in an advanced stage [15].

Despite its good performance for unidirectional laminae, Puck’s theory predicts an initial failure stress of laminates that is significantly lower than experimental results [16]. The underlying reason is that Puck’s theory does not consider the “in situ” effect properly. When confined to a multidirectional laminate, the lamina’s IFF strengths are observed to be higher than those measured in isolated unidirectional laminae due to the constraint of neighboring plies with different fiber orientation [17]. However, directly replacing the strength values in Puck’s IFF criteria with the corresponding in situ strengths may result in incorrect predictions for both the first ply failure load and the fracture angle [18]. Hence Puck’s IFF criteria should be re-examined and modified in order to take into account the in situ effect properly.

Once damage initiates, a stiffness degradation law is required to characterize damage propagation. The direct stiffness degradation method can be easily implemented in a finite element code, but it is purely empirical and lacks generality [19]. The energy-based CDM approach progressively degrades material properties until enough energy is dissipated for complete failure [20]. It associates the damage variables which represent the possible damage modes, with their respective fracture energies. A bilinear or an exponential constitutive relation is often established to describe the failure process. While many progressive damage models of CFRP laminates reduce the stiffness coefficients directly in the material principal coordinate system [21,22,23], a few researchers have noticed the influence of the fracture plane orientation on material degradation [5,24]. As shown in Figure 1, the external stress action plane may not coincide with the fracture plane in the case of inter-fiber failure. Hence, the material principal coordinate system (coordinate 1-2-3) generally differs from the fracture plane coordinate system (coordinate *l*-*n*-*t*). According to Puck’s failure theory, inter-fiber fracture is exclusively determined by the stresses acting on the fracture plane [25]. After damage occurs, the load carrying capacity on the fracture plane will be reduced directly. Therefore, it is more reasonable to define the damage variables in the fracture plane coordinate system instead of the traditional material principal coordinate system.

In the present study, a continuum damage model for intralaminar progressive failure analysis of CFRP laminates is developed based on Puck’s fracture plane theory. The in situ strength effect and the shear nonlinear behavior of CFRPs are considered in the model. The modified Puck’s failure criteria are employed to predict the initiation of fiber and inter-fiber fracture. In strict accordance with Puck’s failure hypothesis, the material stiffness matrix is degraded using the damage variables defined in the fracture plane coordinate system rather than the traditional material principal coordinate system. Theoretical and experimental evaluation validates the rationality and effectiveness of the proposed model. 

## 2. Continuum Damage Model of CFRP Laminates

### 2.1. Stress and Strain Analysis

The intralaminar damage modes of CFRP laminates normally can be divided into two categories: fiber fracture (FF) and inter-fiber fracture (IFF). For the FF mode, the failure plane is approximately the plane on which the longitudinal normal stress *σ*_1_ acts [8], while the IFF mode occurs in an inclined plane parallel to the fiber direction; see Figure 1. The stress and strain components in the fracture plane coordinate system *l*-*n*-*t* can be transformed from those in the material principal coordinate system using the coordinate transformation matrix ***T***:(1)$$\sigma^{fp}=T^{-1}\sigma ,$$
(2)$$\varepsilon^{fp}=T^{T}\varepsilon ,$$
where $\sigma =[\sigma_{1}\;\sigma_{2}\;\sigma_{3}\;\tau_{23}\;\tau_{31}\;\tau_{12}],\;\;\varepsilon =[\varepsilon_{1}\;\varepsilon_{2}\;\varepsilon_{3}\;\gamma_{23}\;\gamma_{31}\;\gamma_{12}],\;\;\sigma^{fp}=[\sigma_{l}\;\sigma_{n}\;\sigma_{t}\;\tau_{nt}\;\tau_{lt}\;\tau_{nl}],\;\;\varepsilon^{fp}=[\varepsilon_{l}\;\varepsilon_{n}\;\varepsilon_{t}\;\gamma_{nt}\;\gamma_{lt}\;\gamma_{nl}]$ are the stress and strain components defined in the coordinate system 1-2-3 and *l*-*n*-*t* respectively. The inverse matrix of ***T*** is given by: (3)$$T^{-1}=\left[\begin{array}{cccccc} 1 & 0 & 0 & 0 & 0 & 0\\ 0 & \cos^{2}\theta & \sin^{2}\theta & 2\sin\theta \cos\theta & 0 & 0\\ 0 & \sin^{2}\theta & \cos^{2}\theta & -2\sin\theta \cos\theta & 0 & 0\\ 0 & -\sin\theta \cos\theta & \sin\theta \cos\theta & \cos^{2}\theta -\sin^{2}\theta & 0 & 0\\ 0 & 0 & 0 & 0 & \cos\theta & -\sin\theta \\ 0 & 0 & 0 & 0 & \sin\theta & \cos\theta \end{array}\right],$$
where *θ* (−90° ≤ *θ* ≤ 90°) is the potential inter-fiber fracture angle.

The constitutive equations for the undamaged unidirectional lamina in the coordinate 1-2-3 and *l*-*n*-*t* are respectively given by:(4)$$\varepsilon =S_{0}\sigma ,$$
(5)$$\varepsilon^{fp}=S_{0}^{fp}\sigma^{fp},$$
where $S_{0}^{}$ and $S_{0}^{fp}$ are the initial compliance matrices defined in the coordinates 1-2-3 and *l*-*n*-*t*, respectively.

Substituting Equations (1) and (2) into Equation (5), we obtain:(6)$$\varepsilon ={\left(T^{T}\right)}^{-1}S_{0}^{fp}T^{-1}\sigma .$$

Comparing Equation (4) and Equation (6) results in:(7)$$S_{0}^{fp}=T^{T}S_{0}T.$$

Once damage initiates, the CDM approach introduces a damage variable matrix ***D*** to establish the relationship between the effective stress $\hat{\sigma }$ and the actual stress ***σ***. As is discussed in Section 1, it is more reasonable to define the effective stress in the fracture plane coordinate system *l*-*n*-*t* instead of the material principal coordinate system 1-2-3:(8)$${\hat{\sigma }}^{fp}=D^{fp}\sigma^{fp},$$
where ***D^fp^*** is the damage variable matrix defined in the coordinate *l*-*n*-*t*, whose form is given by:(9)$$D^{fp}=\mathrm{diag}[\frac{1}{1-d_{l}},\frac{1}{1-d_{n}},\frac{1}{1-d_{t}},\frac{1}{1-d_{nt}},\;\;\;\;\frac{1}{1-d_{lt}},\frac{1}{1-d_{nl}}],$$
where *d_i_* (*i*= *l, n*, *t*) and *d_ij_* (*i*, *j*= *n*, *t*, *l*) are the damage variables corresponding to different damage modes.

Based on the assumption of energy equivalence, the form of the strain energy of the damaged material is identical to that of the undamaged material [26]:(10)$$W_{d}=\frac{1}{2}{\left(\sigma^{fp}\right)}^{T}S_{d}^{fp}\sigma^{fp}=\frac{1}{2}{\left({\hat{\sigma }}^{fp}\right)}^{T}S_{0}^{fp}{\hat{\sigma }}^{fp}\;\;.$$

Substituting Equation (8) into Equation (10) results in:(11)$$\frac{1}{2}{\left(\sigma^{fp}\right)}^{T}S_{d}^{fp}\sigma^{fp}=\frac{1}{2}{\left(\sigma^{fp}\right)}^{T}{\left(D^{fp}\right)}^{T}S_{0}^{fp}D^{fp}\sigma^{fp}.$$

Comparing the left and right sides of Equation (11) gives:(12)$$S_{d}^{fp}\;={\left(D^{fp}\right)}^{T}S_{0}^{fp}D^{fp}.$$

Following the similar deriving process of Equation (7), we obtain:(13)$$S_{d}^{fp}=T^{T}S_{d}T.$$

Substituting Equations (7) and (13) into Equation (12), the compliance matrix of the damaged material in the coordinate 1-2-3, ***S_d_***, is expressed as:(14)$$S_{d}\;={\left(T^{T}\right)}^{-1}{\left(D^{fp}\right)}^{T}T^{T}S_{0}TD^{fp}T^{-1}=D^{T}S_{0}D,$$
where ***D*** is defined as the damage variable matrix in the coordinate 1-2-3:(15)$$D=TD^{fp}T^{-1}.$$

Finally, the constitutive equations for the damaged unidirectional lamina in the coordinate 1-2-3 is given by:(16)$$\varepsilon =S_{d}\sigma .$$

CFRPs usually exhibit nonlinear shear response in the in-plane shear directions. The nonlinear shear constitutive model proposed by Hahn and Tsai [27] is adopted in this paper:(17)$$\left\{ \begin{array}{l} \gamma_{12}=G_{12}^{-1}\tau_{12}+\beta \tau_{12}^{3}\\ \;\gamma_{13}=G_{13}^{-1}\tau_{13}+\beta \tau_{13}^{3} \end{array}\right.,$$
where *G*_12_ and *G*_13_ are the initial shear moduli, and *β* is the shear nonlinearity factor.

In the present study, the nonlinear terms in Equation (17) are linearized using the stable algorithm provided in Abaqus documentation [28]. Although the linearization algorithm does not consider the unloading process, it is adopted due to its simplicity.

### 2.2. Failure Criteria for Intralaminar Damage Initiation

#### 2.2.1. Inter-Fiber Fracture Criteria

According to Puck’s fracture hypothesis, inter-fiber fracture in a plane parallel to the fibers is exclusively determined by the shear stresses *τ_nt_* and *τ_nl_* as well as the normal stress *σ_n_* that are acting on this plane [25]. The stress components *τ_nt_*, *τ_nl_* and *σ_n_* can be calculated from Equation (1):(18)$$\tau_{nt}\left(\theta \right)=-\sigma_{2}\sin\theta \cos\theta +\sigma_{3}\sin\theta \cos\theta +\tau_{23}\left(\cos^{2}\theta -\sin^{2}\theta \right),$$
(19)$$\tau_{nl}\left(\theta \right)=\tau_{31}\sin\theta +\tau_{12}\cos\theta ,$$
(20)$$\sigma_{n}\left(\theta \right)=\sigma_{2}\cos^{2}\theta +\sigma_{3}\sin^{2}\theta +2\tau_{23}\sin\theta \cos\theta .$$

Puck’s failure criteria for inter-fiber fracture are as follows [13] (p. 74):

Inter-fiber tensile fracture (IFFT): (21)$$F_{IFFT}={\left(\frac{\tau_{nt}}{R_{⊥⊥}^{A}}\right)}^{2}+{\left(\frac{\tau_{nl}}{R_{⊥//}^{A}}\right)}^{2}+2\frac{p_{⊥\psi }^{t}}{R_{⊥\psi }^{A}}\sigma_{n}+(1-2\frac{p_{⊥\psi }^{t}R_{⊥}^{At}}{R_{⊥\psi }^{A}})\frac{\sigma_{n}^{2}}{{\left(R_{⊥}^{At}\right)}^{2}}=1\;\;\;\;\;\;\;\;\;\;\mathrm{for}\;\;\;\sigma_{n}\geq 0,$$

Inter-fiber compressive fracture (IFFC):(22)$$F_{IFFC}={\left(\frac{\tau_{nt}}{R_{⊥⊥}^{A}}\right)}^{2}+{\left(\frac{\tau_{nl}}{R_{⊥//}^{A}}\right)}^{2}+2\frac{p_{⊥\psi }^{c}}{R_{⊥\psi }^{A}}\sigma_{n}=1\;\;\;\;\;\;\;\;\;\;\mathrm{for}\;\;\;\;\;\sigma_{n}<0,$$
with
(23)$$\frac{p_{⊥\psi }^{t,c}}{R_{⊥\psi }^{A}}=\frac{p_{⊥⊥}^{t,c}}{R_{⊥⊥}^{A}}\cos^{2}\psi +\frac{p_{⊥//}^{t,c}}{R_{⊥/}^{A}{}_{/}}\sin^{2}\psi ,$$
(24)$$\cos^{2}\psi =\frac{\tau_{nt}^{2}}{\tau_{nt}^{2}+\tau_{nl}^{2}},\;\;\sin^{2}\psi =\frac{\tau_{nl}^{2}}{\tau_{nt}^{2}+\tau_{nl}^{2}},$$
(25)$$R_{⊥}^{At}=Y_{T},$$
(26)$$R_{⊥⊥}^{A}=\frac{Y_{C}}{2\left(1+p_{⊥⊥}^{c}\right)},$$
(27)$$R_{⊥//}^{A}=S_{L}.$$

*Y_T_* and *Y_C_* are the transverse tensile and compressive strengths respectively, while *S_L_* is the longitudinal shear strength. $p_{⊥//}^{t},\;\;p_{⊥//}^{c},\;\;p_{⊥⊥}^{t},\;\;p_{⊥⊥}^{c}$ are four inclination parameters of contour lines of the fracture body. The recommended values of $p_{⊥//}^{t}$ and $p_{⊥//}^{c}$ for CFRPs are 0.35 and 0.30 respectively, while $p_{⊥⊥}^{t}$ and $p_{⊥⊥}^{c}$ can be determined using the following formula [25]:(28)$$p_{⊥⊥}^{t}=p_{⊥⊥}^{c}=\frac{1}{2}\left(\sqrt{1+2p_{⊥//}^{c}\frac{Y_{C}}{S_{L}}}-1\right).$$

Substituting Equations (18)–(20) into Equations (21) and (22), the inter-fiber fracture function, *F_IFF_*, can be written as the function of all stress components except *σ*_1_ as well as the potential fracture angle *θ*:(29)$$F_{IFF}\left(\sigma_{2},\sigma_{3},\tau_{23},\tau_{21},\tau_{31},\theta \right).$$

The *θ*–*F_IFF_* relation can be obtained under any given three-dimensional stress sate. Inter-fiber fracture is considered to occur once the maximum value of *F_IFF_* is equal to 1, and the actual fracture angle *θ_fp_* is the corresponding potential fracture angle:(30)$$F_{IFF}\left(\sigma_{2},\sigma_{3},\tau_{23},\tau_{21},\tau_{31},\theta_{fp}\right)=\underset{-90{}^{\circ}\leq \theta \leq 90{}^{\circ}}{\max}F_{IFF}\left(\sigma_{2},\sigma_{3},\tau_{23},\tau_{21},\tau_{31},\theta \right)=1.$$

The maximum value of *F_IFF_* and the corresponding fracture angle *θ_fp_* can be searched numerically within the interval −90° ≤ *θ* ≤ 90° [17].

Given the in situ effect of multidirectional laminates, the in situ strength values, $Y_{T}^{is}$, $Y_{C}^{is}$, and $S_{L}^{is}$, should be used in Puck’s IFF criteria. However, simply replacing the strength values in Equations (25)–(28) with the corresponding in situ strengths may result in incorrect predictions of both the failure stress and the corresponding fracture angle [18]. For example, the strength values of CFRP IM7/8552 are listed in Table 1. The following simple stress states are considered: transverse tensile stress ($\sigma_{2}=Y_{T}$ or $Y_{T}^{is}$), transverse compressive stress ($\sigma_{2}=-Y_{C}$ or $-Y_{C}^{is}$), and longitudinal shear stress ($\tau_{21}=S_{L}$ or $S_{L}^{is}$). As shown in Figure 2, for the ply in a unidirectional laminate, the maximum values of the failure function *F_IFF_* and the corresponding fracture angles are well predicted under all these simple stress states. For the ply embedded in a multidirectional laminate, Figure 3 shows that the predicted results are reasonable under transverse compression (see Appendix A for details) and longitudinal shear. Nevertheless, the maximum value of *F_IFF_* exceeds 1 under pure transverse tension, indicating that material failure has already occurred before $\sigma_{2}=Y_{T}^{is}$. Besides, the corresponding tensile fracture angle is no longer 0°, which is obviously wrong [18]. To reasonably combine Puck’s IFF criteria with the in situ strength theory, the procedure to determine the seven unknown parameters, namely, the three strength parameters ($R_{⊥}^{At}$, $R_{⊥⊥}^{A}$ and $R_{⊥//}^{A}$) and the four inclination parameters ($p_{⊥//}^{t}$, $p_{⊥//}^{c}$, $p_{⊥⊥}^{t}$ and $p_{⊥⊥}^{c}$), should be re-examined in detail.

Under the longitudinal shear stress state, the corresponding in situ strength is denoted as $S_{L}^{is}$, and the fracture angle $\theta_{fp}^{ls}=0{}^{\circ}$. According to Equations (18)–(20), the only stress component on the fracture plane is *τ_nl_* = $S_{L}^{is}$. Substituting (*τ_nt_* = 0, *τ_nl_* = $S_{L}^{is}$, *σ_n_* = 0) into Equation (21) results in:(31)$$R_{⊥//}^{A}=\;S_{L}^{is}.$$

Under transverse compression, the corresponding in situ strength is denoted as $Y_{C}^{is}$. The compressive fracture angle is slightly above 50° for CFRPs [25]; thus, $\theta_{fp}^{c}=51{}^{\circ}\;$ is taken in the present study. According to Equations (18)–(20), the stress components on the fracture plane are *τ_nt_* = $Y_{C}^{is}\sin\theta_{fp}^{c}\cos\theta_{fp}^{c}$, *σ_n_* = $-Y_{C}^{is}\cos^{2}\theta_{fp}^{c}$. Substituting (*τ_nt_* = $Y_{C}^{is}\sin\theta_{fp}^{c}\cos\theta_{fp}^{c}$, *τ_nl_* = 0, *σ_n_* = $-Y_{C}^{is}\cos^{2}\theta_{fp}^{c}$) into Equation (22) results in:(32)$${\left(\frac{Y_{C}^{is}\sin\theta \cos\theta }{R_{⊥⊥}^{A}}\right)}^{2}-2\frac{p_{⊥⊥}^{c}}{R_{⊥⊥}^{A}}Y_{C}^{is}\cos^{2}\theta =1.$$

$\theta_{fp}^{c}$ is the fracture angle corresponding to the maximum value of the failure function *F_IFFC_*:(33)$${\frac{dF_{IFFC}\left(\theta \right)}{d\theta }|}_{\theta =\theta_{fp}^{c}}={\frac{d}{d\theta }\left[{\left(\frac{Y_{C}^{is}\sin\theta \cos\theta }{R_{⊥⊥}^{A}}\right)}^{2}-2\frac{p_{⊥⊥}^{c}}{R_{⊥⊥}^{A}}Y_{C}^{is}\cos^{2}\theta \right]|}_{\theta =\theta_{fp}^{c}}=0\;.$$

Solving Equations (32) and (33) together, we obtain:(34)$$R_{⊥⊥}^{A}=\frac{Y_{C}^{is}}{2\left(1+p_{⊥⊥}^{c}\right)},$$
(35)$$p_{⊥⊥}^{c}=\frac{1}{2\cos^{2}\theta_{fp}^{c}}-1.$$

Under transverse tension, the corresponding in situ strength is denoted as $Y_{T}^{is}$, and the fracture angle $\theta_{fp}^{t}=0{}^{\circ}$. According to Equations (18)–(20), the only stress component on the fracture plane is *σ_n_* = $Y_{T}^{is}$. Substituting (*τ_nt_* = 0, *τ_nl_* = 0, *σ_n_* = $Y_{T}^{is}$) into Equation (21) results in:(36)$$R_{⊥}^{At}=Y_{T}^{is}.$$

The inclination parameters $p_{⊥//}^{t}\;$ and $p_{⊥//}^{c}$ can be derived experimentally [25]. Nevertheless, the inclination parameters $p_{⊥⊥}^{t}$ and $p_{⊥⊥}^{c}$ in Puck’s IFF criteria are not determined by experiments. Their recommended values, Equation (28), are given based on mathematical reasons, rather than physical ones [25]. If the in situ strengths are taken into account, Equation (28) will lead to incorrect predictions under pure transverse tension (as shown in Figure 3). In the present study, the rigorously derived Equation (35) is employed to determine $p_{⊥⊥}^{c}$, while $p_{⊥⊥}^{t}$ is obtained using the following method.

According to Equations (18)–(20), the stress components on the potential fracture plane under pure transverse tensile stress (*σ*_2_ > 0) are given by:(37)$$\tau_{nt}\left(\theta \right)=-\sigma_{2}\sin\theta \cos\theta ,\;\tau_{nl}\left(\theta \right)=0,\;\sigma_{n}\left(\theta \right)=\sigma_{2}\cos^{2}\theta .$$

The failure stress is $Y_{T}^{is}$, and the fracture angle $\theta_{fp}^{t}=0{}^{\circ}$. This implies the following conditions:(38)$${F_{IFFT}\left(\theta \right)|}_{\theta =0^{\circ },\;\tau_{nt}=0,\;\tau_{nl}=0,\sigma_{n}=Y_{T}^{is}}=1,$$
(39)$${F_{IFFT}\left(\theta \right)|}_{\;\tau_{nt}=-Y_{T}^{is}\sin\theta \cos\theta ,\;\tau_{nl}=0,\sigma_{n}=Y_{T}^{is}\cos^{2}\theta }\leq 1\;\;\;\;\;\;\;\;\;\mathrm{for}\;\;\;\;\;\theta \in [-90{}^{\circ},90{}^{\circ}].$$

It is easy to prove that Equation (38) can be satisfied. From Equation (39) we obtain:(40)$${\left(\frac{-Y_{T}^{is}\sin\theta \cos\theta }{R_{⊥⊥}^{A}}\right)}^{2}+2\frac{p_{⊥⊥}^{t}}{R_{⊥⊥}^{A}}Y_{T}^{is}\cos^{2}\theta +(1-2\frac{p_{⊥⊥}^{t}R_{⊥}^{At}}{R_{⊥⊥}^{A}})\frac{{\left(Y_{T}^{is}\cos^{2}\theta \right)}^{2}}{{\left(R_{⊥}^{At}\right)}^{2}}\leq 1\;\;\;\;\;\;\mathrm{for}\;\;\;\;\;\theta \in [-90{}^{\circ},90{}^{\circ}].$$

Substituting Equation (36) into Equation (40) results in:(41)$$p_{⊥⊥}^{t}\leq \frac{\left[1-{\left(\frac{Y_{T}^{is}\sin\theta \cos\theta }{R_{⊥⊥}^{A}}\right)}^{2}-\cos^{4}\theta \right]R_{⊥⊥}^{A}}{2Y_{T}^{is}\cos^{2}\theta \left(1-\cos^{2}\theta \right)}\;\;\;\;\;\;\;\;\;\;\;\;\;\;\;\;\mathrm{for}\;\;\;\;\;\theta \in [-90{}^{\circ},90{}^{\circ}].$$

Puck et al. believed that $p_{⊥⊥}^{t}$ and $p_{⊥⊥}^{c}$ ought to be approximately of the same magnitude, and setting $p_{⊥⊥}^{t}=\;p_{⊥⊥}^{c}$ will not lead to any unacceptable contradictions for unidirectional laminates [25]. Since the modified IFF criteria should also be applicable in the case of unidirectional laminates, we propose to minimize the difference between $p_{⊥⊥}^{t}$ and $p_{⊥⊥}^{c}$ without violation of Equation (41). In other words, $p_{⊥⊥}^{t}$ can be obtained by solving the following problem: (42)$$\left\{ \begin{array}{l} \mathrm{find}\;\;\;\;\;\;\;\;\;\;\;\;\;p_{⊥⊥}^{t}\\ s.t.\;\;\;\;\;\;\;\;\;\;\;\;\;p_{⊥⊥}^{t}\leq \frac{\left[1-{\left(\frac{Y_{T}^{is}\sin\theta \cos\theta }{R_{⊥⊥}^{A}}\right)}^{2}-\cos^{4}\theta \right]R_{⊥⊥}^{A}}{2Y_{T}^{is}\cos^{2}\theta \left(1-\cos^{2}\theta \right)}\;\;\;\;\;\;\;\;\;\;\;\;\;\;\;\;\mathrm{for}\;\;\;\;\;\theta \in [-90{}^{\circ},90{}^{\circ}]\\ \min\;\;\;\;\;\;\;\;\;\;\;\;\left|p_{⊥⊥}^{t}\;-p_{⊥⊥}^{c}\right| \end{array}\right.$$

#### 2.2.2. Fiber Fracture Criteria

Puck suggested the use of a simple maximum stress formulation to predict fiber fracture in 1969, and believed it was sufficient for a preliminary analysis [13] (p. 37). More sophisticated FF criteria were developed afterwards to account for the transverse effect, requiring the measurements for the elastic modulus *E*_1*f*_ and Poisson’s ratio *υ*_12*f*_ of the fibers [17]. Since these values are seldom provided in references, the simple maximum stress criteria are adopted in the present study:

Fiber tensile fracture (FFT): (43)$$F_{FFT}=\frac{\sigma_{1}}{X_{T}}=1\;\;\;\;\;\;\;\;\;\;\;\mathrm{for}\;\;\;\sigma_{1}\geq 0,$$

Fiber compressive fracture (FFC):(44)$$F_{FFC}=\frac{\sigma_{1}}{X_{C}}=1\;\;\;\;\;\;\;\;\;\;\;\;\mathrm{for}\;\;\;\sigma_{1}<0,$$
where *X_T_* and *X_C_* are the longitudinal tensile and compressive strengths, respectively.

### 2.3. Damage Evolution Law

Damage propagation is a process accompanied by the dissipation of energy. A fracture energy-based approach is employed in the present study to characterize damage evolution. Once damage initiation is predicted by the failure criteria, the material stiffness will be progressively degraded until enough energy is dissipated for complete failure.

#### 2.3.1. Damage Variables

As shown in Figure 4, a bilinear constitutive relation is used to characterize the damage evolution process. The internal damage variable corresponding to each intralaminar failure mode is defined as:(45)$$d_{I}=\max\left\{0,\min\left\{1,\frac{\varepsilon_{eq,I}^{f}\left(\varepsilon_{eq,I}-\varepsilon_{eq,I}^{0}\right)}{\varepsilon_{eq,I}\left(\varepsilon_{eq,I}^{f}-\varepsilon_{eq,I}^{0}\right)}\right\}\right\}\;\;\;\;\;\;\;\;I\in \left(IFFT,IFFC,FFT,FFC\right),$$
where $\varepsilon_{eq,I}$, $\varepsilon_{eq,I}^{0}$ and $\varepsilon_{eq,I}^{f}$ are the equivalent strains in the current state, the damage initiation state, and the final failure state, respectively. All these equivalent strains will be defined precisely later. *d_I_* = 0 means the material is undamaged, while *d_I_* = 1 represents the complete failure of material.

The normal tensile stress on the inter-fiber fracture plane (*σ_n_* > 0) tends to open the cracks, so no forces can be transmitted as a result of there being no contact between the crack faces. Nevertheless, cracks are closed under the normal compressive stress (*σ_n_* < 0), and thus forces can be transmitted across the cracks [5]. Consequently, the damage variable $d_{IFFC}$ only has an effect on the shear moduli, and the damage variable with respect to the *n*-direction is given by:(46)$$d_{n}=d_{IFFT}.$$

The damage variable with respect to the *l*-direction is given by [21]:(47)$$d_{l}=d_{FFT}+d_{FFC}-d_{FFT}d_{FFC}.$$

The damage variable *d_t_* represents the damage with respect to the *t*-direction. Since the *t*-direction is always perpendicular to the damage axes of IFF and FF failure (see Figure 1), no damage occurs along the *t*-axis [31], i.e.,
(48)$$d_{t}=0.$$

The damage variables associated with the shear moduli *G_ij_* (*i*, *j* = *n*, *t*, *l*) can be expressed as:(49)$$d_{nt}=1-(1-d_{m})(1-d_{t})=d_{m},$$
(50)$$d_{lt}=1-(1-d_{l})(1-d_{t})=d_{l},$$
(51)$$d_{nl}=1-(1-d_{m})(1-d_{l})=d_{l}+d_{m}-d_{l}d_{m},$$
with
(52)$$d_{m}=d_{IFFT}+d_{IFFC}-d_{IFFT}d_{IFFC}.$$

#### 2.3.2. Equivalent Stress and Strain

According to Puck’s fracture hypothesis, material failure is exclusively determined by the stress components acting on the fracture plane. For the IFF mode, the stress components on the fracture plane are *τ_nt_*, *τ_nl_* and *σ_n_*. The tensile normal stress (*σ_n_* > 0) promotes fracture in combination with the shear stresses *τ_nt_* and *τ_nl_*, while the compressive normal stress (*σ_n_* < 0) impedes material failure [25]. Therefore, the equivalent stress and strain for the IFF mode are defined as:(53)$$\sigma_{\mathrm{eq},\mathrm{IFF}}=\sqrt{{\langle \sigma_{n}\rangle }^{2}+\tau_{nt}^{2}+\tau_{nl}^{2}},$$
(54)$$\varepsilon_{\mathrm{eq},\mathrm{IFF}}=\sqrt{\;{\langle \varepsilon_{n}\rangle }^{2}+\gamma_{nt}^{2}+\gamma_{nl}^{2}},$$
where 〈    〉 is the McCauley operator defined as 〈x〉=(x+|x|)/2 for x∈ℜ.

For the FF mode, the stress components on the fracture plane are *τ*_12_, *τ*_13_ and *σ*_1_ [8]. However, the contribution of the shear stress components *τ*_12_ and *τ*_13_ to fiber fracture is very small, and hence can be neglected [32]. Therefore, the equivalent stress and strain for the FF mode are defined as:(55)$$\sigma_{\mathrm{eq},\mathrm{FF}}=\sqrt{\sigma_{1}^{2}},$$
(56)$$\varepsilon_{\mathrm{eq},\mathrm{FF}}=\sqrt{\varepsilon_{1}^{2}}.$$

The equivalent stress and strain in the damage initiation state, $\sigma_{eq,I}^{0}\;$ and $\varepsilon_{eq,I}^{0}$, correspond to the equivalent stress and strain when the failure function *F_I_* equals 1.

#### 2.3.3. Equivalent Strain in the Final Failure State

The crack band theory introduces the characteristic length $L^{c}$ to correlate the energy release rate (i.e., energy dissipated per unit area) with the energy dissipated per unit volume [33]. For the IFF failure mode, a quadratic interaction criterion is established under mixed-mode loading:(57)$${\left(\frac{g_{n}}{G_{Ic}/L_{IFF}^{c}}\right)}^{2}+{\left(\frac{g_{nt}}{G_{IIc}/L_{IFF}^{c}}\right)}^{2}+{\left(\frac{g_{nl}}{G_{IIc}/L_{IFF}^{c}}\right)}^{2}=1,$$
where *g_n_*, *g_nt_*, and *g_nl_* are the strain energy densities associated with the corresponding stress components. *G_Ic_* and *G_IIc_* are the critical energy release rates for fracture mode I and mode II respectively. $L_{IFF}^{c}$ is the characteristic length for IFF.

The strain energy densities in the final IFF failure state are given by [12]:(58)$$g_{i}^{f}=\int_{0}^{\varepsilon_{i}^{f}}\sigma_{i}d\varepsilon_{i}\approx \frac{1}{2}\sigma_{i}^{0}\varepsilon_{i}^{f}=\frac{1}{2}\sigma_{i}^{0}\beta_{i}\varepsilon_{eq}^{f}\;\;\;\;\;\;\;(i=n,nt,nl),$$
where $\sigma_{i}^{0}$ is the stress at the initiation of IFF failure. $\beta_{i}$ denotes the mixed-mode ratio, which can be expressed as:(59)$$\beta_{n}=\frac{\langle \varepsilon_{n}\rangle }{\varepsilon_{eq}},\;\beta_{nt}=\frac{\gamma_{nt}}{\varepsilon_{eq}}\;\;,\;\beta_{nl}=\frac{\gamma_{nl}}{\varepsilon_{eq}}\;.$$

Substituting Equation (58) into Equation (57), the equivalent strain in the final IFF failure state is given by:(60)εeq,IFFf=2LIFFc [(σn0βnGIc)2+(τnt0βntGIIc)2+(τnl0βnlGIIc)2]−12.

For the FF mode, the equivalent strain in the final failure state is determined by the bilinear constitutive relation shown in Figure 4:(61)$$\frac{G_{fc}^{t\left(c\right)}}{L_{FF}^{c}}=\frac{1}{2}X_{T\left(C\right)}\varepsilon_{eq,FF}^{f}\Rightarrow \varepsilon_{eq,FF}^{f}=\frac{2G_{fc}^{t\left(c\right)}}{X_{T\left(C\right)}L_{FF}^{c}}\;,$$
where $G_{fc}^{t\left(c\right)}$ is the critical energy release rate corresponding to longitudinal tension (compression). *X_T_*_(*C*)_ is the longitudinal tensile (compressive) strength. $L_{FF}^{c}$ is the characteristic length for FF.

## 3. Theoretical and Experimental Verification

### 3.1. Ply Failure Analysis

Two plies, one in a unidirectional laminate and the other embedded in a multidirectional laminate, are taken as the case study. The material system is IM7/8552, whose strength values are listed in Table 1. 

As shown in Figure 5, the results predicted by the original and modified Puck’s IFF criteria are almost identical for the ply in a unidirectional laminate. There are only negligible discrepancies between the curves due to the small differences between the parameters used ($p_{⊥⊥}^{t}=\;p_{⊥⊥}^{c}=0.258$ and 0.262 respectively). The failure stresses and the fracture angles are correctly predicted under these simple stress states.

For the embedded lamina, the predicted results are reasonable under transverse compression (see Appendix A for details) and longitudinal shear; see Figure 6b,c. Under pure transverse tension, the failure stress is underestimated by the original IFF criteria, and the predicted fracture angle is unreasonable; see Figure 6a. This problem is solved by using the modified IFF criteria.

For the ply in a unidirectional laminate, again there is a good agreement between the predicted *σ*_2_-*τ*_21_ failure envelopes; see Figure 7a. For the embedded lamina, Puck’s original IFF criteria underestimate the failure stress in the high tensile stress (*σ*_2_ > 0) region (see Figure 7b), while the modified criteria do not have this serious defect.

### 3.2. Progressive Failure Analysis of CFRP Laminates

The proposed model focuses on the intralaminar damage of CFRP laminates, yet ignores the interlaminar damage between adjacent layers (i.e., delamination). Since the influence of delamination on the mechanical behavior of composite laminates with dispersed plies is very small under in-plane loading conditions [29,34], notched CFRP laminates under uniaxial tension and compression are selected as the validation cases. To avoid convergence problems, quasi-static analysis was performed in the FE software Abaqus/Explicit using a user-defined material subroutine (VUMAT). The FE model of the specimen is shown in Figure 8. The continuum 3D 8-node reduced integration element (C3D8R) was used per ply thickness. The mesh around the circular notch was refined (0.5 mm × 0.5 mm × *t*), while a relatively coarse mesh was used in the remaining regions (1.5 mm × 0.5 mm × *t*). Ideally, the characteristic length is a function of the fracture angle as well as the direction of crack propagation. For the sake of simplicity, the characteristic lengths for IFF and FF are here taken as the cubic root of the volume of the elements. This method has been proven to be efficient and effective for 3D elements if the element size is sufficiently small [24,35].

Only half of the laminate was modelled due to the stacking symmetry in the *z* direction. The loading direction is along the longitudinal direction of the specimen, which coincides with the direction of the 0° ply. The laminate was clamped on the left end, while a uniform displacement was applied at the right edge. The *z*-symmetry boundary conditions were applied at the mid-plane.

Two specimens were modelled in the present study. One is a quasi-isotropic laminate with the stacking sequence [90/0/±45]_3s_ under unidirectional tensile loading [29], while the other is an angle-ply laminate containing six ±45° sub-laminates under uniaxial compression [36]. A sketch of the laminate is shown in Figure 9, and the geometric dimensions are reported in Table 2. A strain gauge was placed on the outer surface (*l_gau_* = 50 mm) to monitor the axial strain *ε* of the quasi-isotropic laminate, while an extensometer was used to measure the relative displacement Δ of the angle-ply laminate (*l_ext_* = 25.4 mm). The laminates are manufactured from CFRP IM7/8552 and T300/976 respectively, whose mechanical properties are listed in Table 3 and Table 4. The in situ strengths are calculated using the analytical formulas proposed by Camanho and his co-workers [30,37]; see Table 5.

For the laminate under uniaxial tensile loading, a total of 110,784 elements were used in the model. The axial average stress is defined as the external load per unit cross-sectional area: *σ* = *F*/*wt_L_*, where *w* and *t_L_* are the width and thickness of the specimen. The axial average stress–strain curves are shown and compared in Figure 10. Good correlation between the experimental and numerical results is observed, and the predicted ultimate average stress (394.8 MPa) is slightly higher than the experimental value (375.7 MPa). Not considering interlaminar damage might be another possible reason for the overprediction apart from the uncertainty of experiments. As shown in Figure 11a, inter-fiber damage first occurs at the notch edge of the outer 90° ply. Subsequently, fiber tensile fracture initiates in the 0° plies; see Figure 11b. As the load increases, damage propagates perpendicular to the loading direction. In the final failure state of the laminate, fiber damage extends along the transverse direction in the 0° plies, while inter-fiber damage in the 90° plies fully extends across the width of the specimen; see Figure 11c,d. Inter-fiber tensile fracture also occurs in the ±45° plies (see Figure 11e,f), but the damage zones are smaller than those in the 90° plies. The predicted damage pattern is in line with the net-section failure mode observed in the experiment [29].

For the laminate under uniaxial compressive loading, a total of 38,892 elements were used in the model. The *F*–Δ curve obtained from the experimental data is plotted in Figure 12, where *F* is the external load measured in the test, and Δ is the relative displacement. As shown in Figure 12, the curve exhibits a very pronounced nonlinear behavior, and the ultimate failure load is approximately 13.5 kN. The simulated curve agrees well with the experimental result, and the ultimate failure load is well predicted (12.6 kN) with a relative error of 6.7%. No fiber damage is observed in the whole failure process of the laminate. Only inter-fiber compressive damage occurs in the vicinity of the hole, and then propagates along the ±45° direction; see Figure 13. The predicted damage pattern is consistent with the experimental observation [36].

## 4. Conclusions

In the present study, a continuum damage model based on the modified Puck’s theory is developed to simulate the intralaminar progressive failure of CFRP laminates. The in situ strength effect and the nonlinear shear behavior of CFRPs are considered in the model. The modified Puck’s failure criteria are adopted to determine damage initiation, while a bilinear constitutive relation is used to describe damage evolution. In strict accordance with Puck’s concept of action plane, the equivalent stress/strain and the damage variables are defined in the fracture plane coordinate system rather than the traditional material principal coordinate system. Theoretical and experimental evaluation of CFRP laminates validates the rationality and effectiveness of the proposed model. The numerical model has been implemented in an FE software to simulate the progressive failure of CFRP laminates, and good correlation between the numerical and experimental results is observed. Future research will combine the proposed model with the interface fracture modeling techniques to simulate both intralaminar and interlaminar damage of CFRP laminates.

## Figures and Tables

**Figure 1 materials-12-03292-f001:**
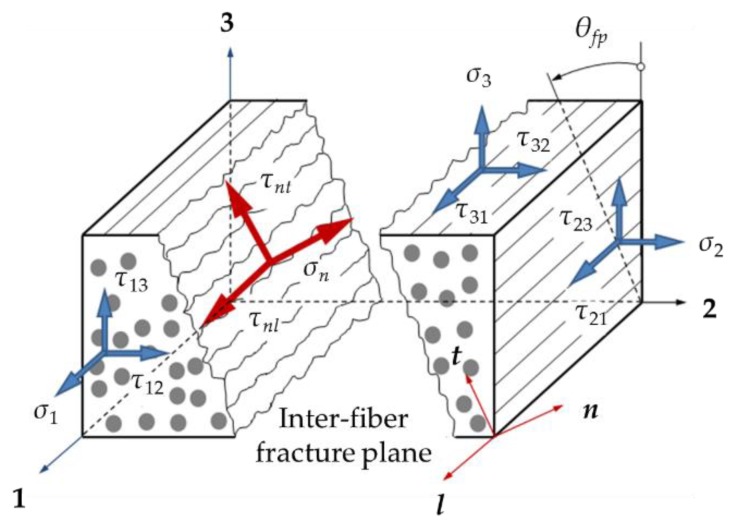
Definition of the material principal coordinate system, 1-2-3, and the fracture plane coordinate system, *l*-*n*-*t*.

**Figure 2 materials-12-03292-f002:**
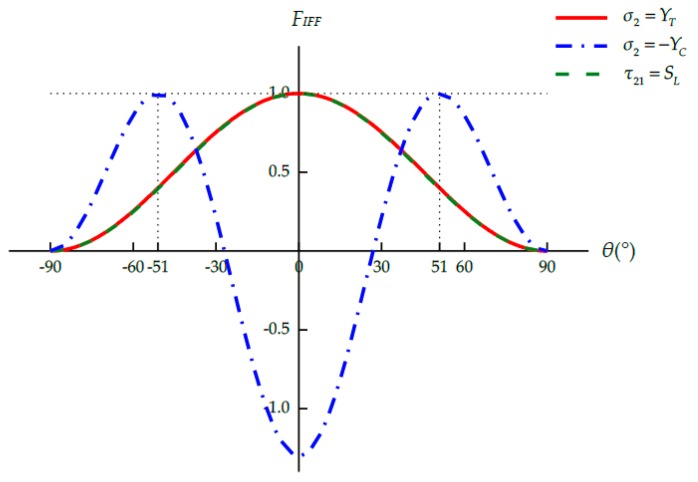
*F_IFF_*–*θ* relation under simple stress states for the ply in a unidirectional laminate.

**Figure 3 materials-12-03292-f003:**
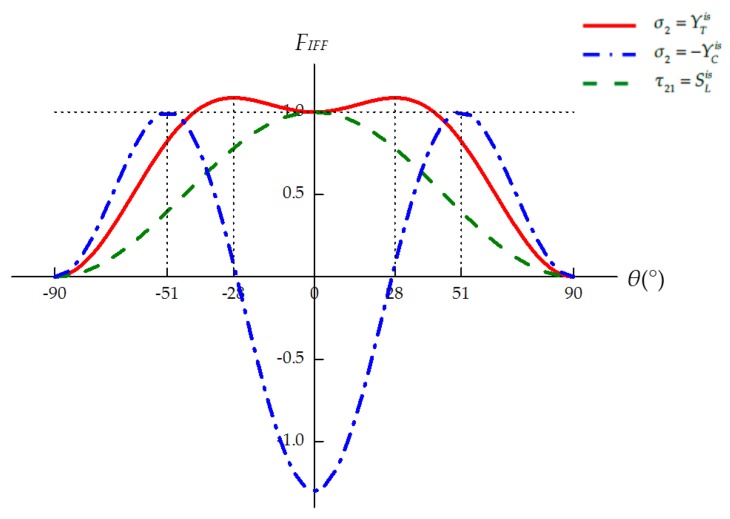
*F_IFF_*–*θ* relation under simple stress states for the ply embedded in a multidirectional laminate.

**Figure 4 materials-12-03292-f004:**
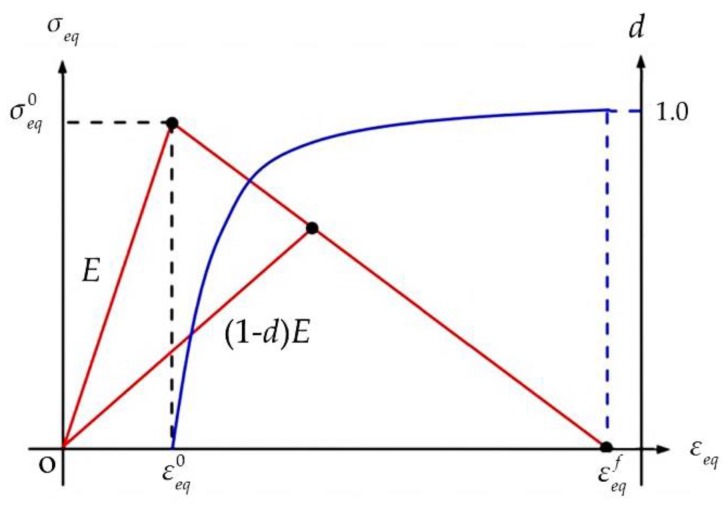
Bilinear constitutive relation. *E* is the elastic modulus of the undamaged material.

**Figure 5 materials-12-03292-f005:**
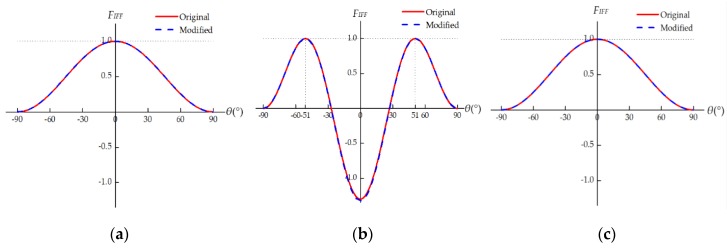
*F_IFF_*–*θ* relation under simple stress states for the ply in a unidirectional laminate. (**a**) $\sigma_{2}=Y_{T}$; (**b**)$\sigma_{2}=-Y_{C}$; (**c**)$\tau_{21}=S_{L}$.

**Figure 6 materials-12-03292-f006:**
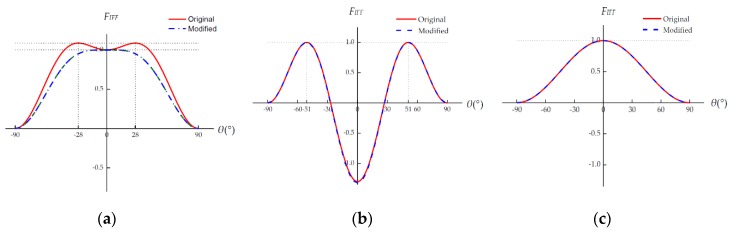
*F_IFF_*–*θ* relation under simple stress states for the ply embedded in a multidirectional laminate. (**a**)$\sigma_{2}=Y_{T}^{is}$; (**b**)$\sigma_{2}=-\;Y_{C}^{is}$; (**c**)$\tau_{21}=S_{L}^{is}$.

**Figure 7 materials-12-03292-f007:**
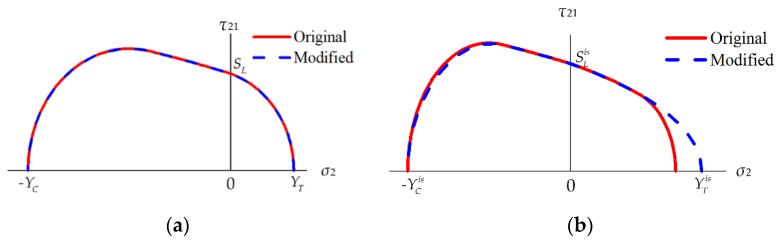
*σ*_2_-*τ*_21_ failure envelopes. (**a**) Ply in a unidirectional laminate; (**b**) Ply embedded in a multidirectional laminate.

**Figure 8 materials-12-03292-f008:**
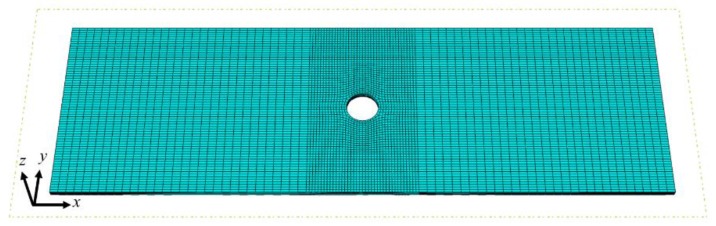
The FE model of the specimen.

**Figure 9 materials-12-03292-f009:**
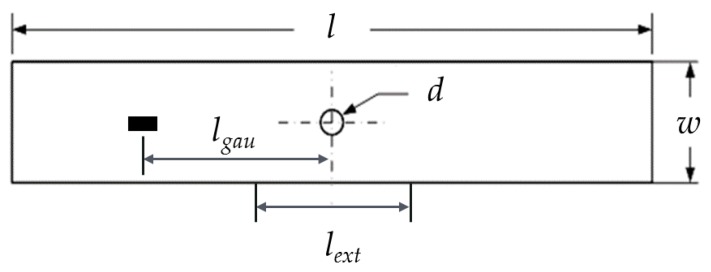
A sketch of the laminate.

**Figure 10 materials-12-03292-f010:**
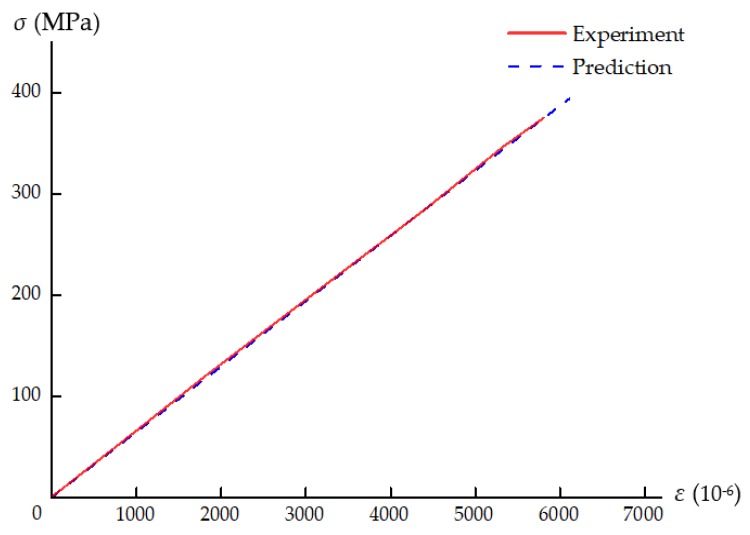
Experimental and numerically predicted axial average stress–strain curves.

**Figure 11 materials-12-03292-f011:**
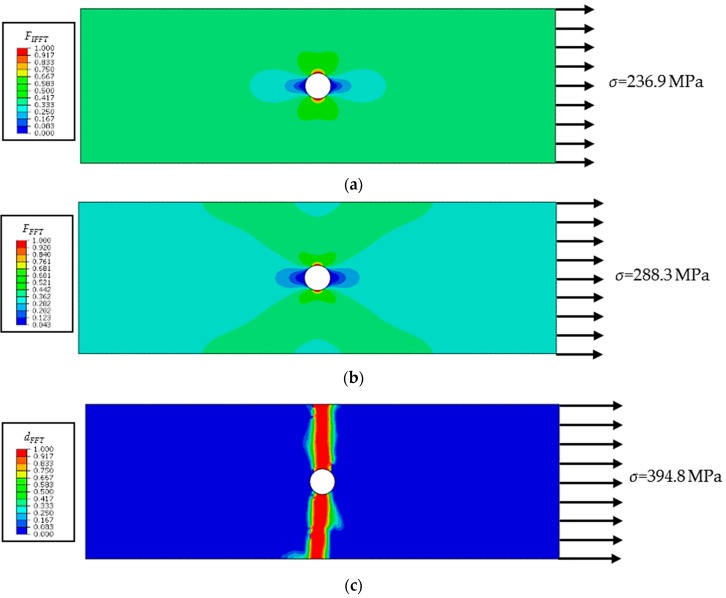
Progressive failure process of the CFRP laminate under tension. (**a**) IFFT initiates in a 90° ply; (**b**) FFT initiates in a 0° ply; (**c**) Damage variable *d_FFT_* in a 0° ply; (**d**) Damage variable *d_IFFT_* in a 90° ply; (**e**) Damage variable *d_IFFT_* in a 45° ply; (**f**) Damage variable *d_IFFT_* in a −45° ply.

**Figure 12 materials-12-03292-f012:**
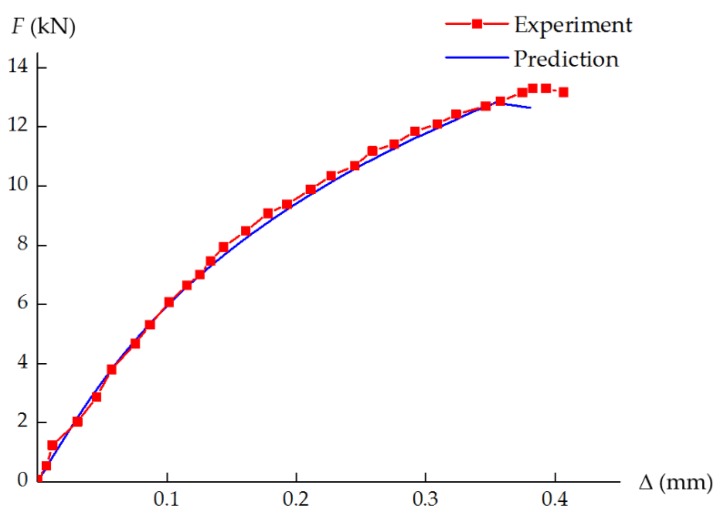
Experimental and numerically predicted *F*–Δ curves

**Figure 13 materials-12-03292-f013:**
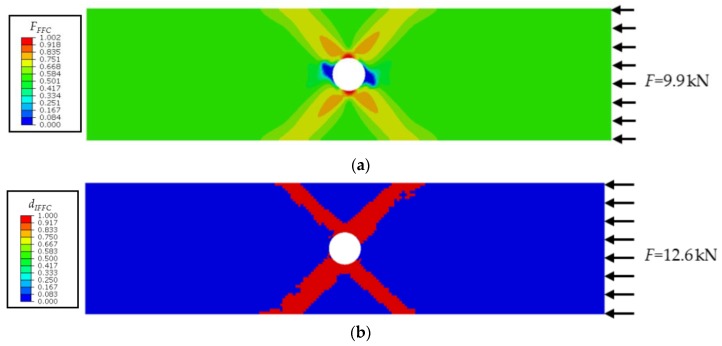
Progressive failure process of the CFRP laminate under compression. (**a**) IFFC initiates in a 45° ply; (**b**) Damage variable *d_IFFC_* in a 45° ply.

**Table 1 materials-12-03292-t001:** Strength values of CFRP IM7/8552 [29].

*Y_T_*/MPa	*Y_C_*/MPa	*S_L_*/MPa	$Y_{T}^{\mathrm{is}}$/MPa	$Y_{C}^{\mathrm{is}}$/MPa	$S_{L}^{\mathrm{is}}$/MPa
62.3	199.8	92.3	160.2	281.8 ^1^	130.2

^1^$Y_{C}^{is}$ is calculated using $Y_{C}^{is}=\frac{S_{L}^{is}}{S_{L}}Y_{C}$ in reference [30].

**Table 2 materials-12-03292-t002:** Geometric dimensions of the laminates.

Stacking Sequence	*l* (mm)	*w* (mm)	*d* (mm)
[90/0/±45]_3s_	150	48	8
[±45]_6s_	101.6	25.4	6.35

**Table 3 materials-12-03292-t003:** Mechanical properties of material IM7/8552 [29].

***E*_1_/GPa**	***E*_2_/GPa**	***G*_12_/GPa**	***υ*_12_**	***tp*/*mm***
171.42	9.08	5.29	0.32	0.131
***X_T_*/MPa**	***X_C_*/MPa**	***Y_T_*/MPa**	***Y_C_*/MPa**	***S_L_*/MPa**
2326.2	1200.1	62.3	199.8	92.3
***G_Ic_*/kJm^−2^**	***G_IIc_*/kJm^−2^**	$\;G_{fc}^{t}$ **/kJm^−2^**	$\;G_{fc}^{c}$ **/kJm^−2^**	***β*/MPa^−3^**
0.2774	0.7879	81.5	106.3	2.98 × 10^−8^

**Table 4 materials-12-03292-t004:** Mechanical properties of material T300/976 [36,38].

***E*_1_/GPa**	***E*_2_/GPa**	***G*_12_/GPa**	***υ*_12_**	***tp*/*mm***
156.5	12.9	6.96	0.23	0.143
***X_T_*/MPa**	***X_C_*/MPa**	***Y_T_*/MPa**	***Y_C_*/MPa**	***S_L_*/MPa**
1516.8	1592.7	44.54	253	106.8
***G_Ic_*/kJm^−2^**	***G_IIc_*/kJm^−2^**	$\;G_{fc}^{t}$ **/kJm^−2^**	$\;G_{fc}^{c}$ **/kJm^−2^**	***β*/MPa^−3^**
0.22	0.46	91.6	79.9	2.44 × 10^−8^

**Table 5 materials-12-03292-t005:** Analytical formulas to calculate the in situ strengths.

Type of Ply	$Y_{T}^{\mathrm{is}}$	$S_{L}^{\mathrm{is}}$	$Y_{C}^{\mathrm{is}}$
Thin Embedded Ply	$\sqrt{\frac{8G_{Ic}}{\pi t\Lambda_{22}^{0}}},\;\Lambda_{22}^{0}=2\left(\frac{1}{E_{2}}-\frac{\nu_{12}^{2}}{E_{1}}\right)$	$\sqrt{\frac{{\left(1+\beta \varphi G_{12}^{2}\right)}^{1/2}-1}{3\beta G_{12}}},\;\varphi =\frac{48G_{IIc}}{\pi t}$	$\frac{S_{L}^{is}}{S_{L}}Y_{C}$
Thin Outer Ply	$1.78\sqrt{\frac{G_{Ic}}{\pi t\Lambda_{22}^{0}}},\;\Lambda_{22}^{0}=2\left(\frac{1}{E_{2}}-\frac{\nu_{12}^{2}}{E_{1}}\right)$	$\sqrt{\frac{{\left(1+\beta \varphi G_{12}^{2}\right)}^{1/2}-1}{3\beta G_{12}}},\;\varphi =\frac{24G_{IIc}}{\pi t}$	$\frac{S_{L}^{is}}{S_{L}}Y_{C}$

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
