# Peer review of "A Continuum Damage Model for Intralaminar Progressive Failure Analysis of CFRP Laminates Based on the Modified Puck’s Theory"

_materials, 2019, doi:10.3390/ma12203292_

Round 1
Reviewer 1 Report
The paper deals with a modification of the Puck's model to better taking into account the damage initiation in the embedded plies. Puck's inter-fiber failure criteria have been modified to take into account the in-situ effects.
Comments :
page 2 line 57 : define IFF and FF
page 8 line 220: precise that the equivalent strain will be defined later
page 13 : Can you precise the number of elements and their size. What is the value of the characteristic lengths (FF and IFF)
page 13 : A very good agreement with experimental data has been obtained for the 2 examples with modified criteria. Did the original ones give also good results? Are the selected examples are significant to underline the efficiency of the "new model"?
figures 11 and 13 : It would be good to show the evolution of the damages during the loading and to correlate stress-strain curves (figures 10 and 12)
figures 12 and 13 : Can you define the stress? is it the nominal one?
Why did the authors not use an example with loading-unloading applied to a cross-ply laminate to show the evolution of the damage? Without plasticity criterion the residual strain after damage occurrence can be predicted with the model based on CDM.
Why did the authors define a 3D model (maybe to use continuum shell element) and apply it in the case of plane stress (2D) problem?
Reviewer 2 Report
This manuscript presents a modification to a well-established set of failure criteria for unidirectional fibre-reinforced composite materials to account for thickness effects on the strength of unidirectional layers embedded in multidirectional laminates, and implementation of the proposed modified criteria in combination with a continuum damage mechanics models in a finite element context. The proposed modification is pertinent, and therefore worth publication. However, the reported analysis shows some issues that require mandatory revision.
Major issues:
Lines 163-164 and 274: it is mentioned that “for the ply embedded in a multidirectional laminate, (…) the predicted results are correct (reliable) under transverse compression”. However, this is only correct in this case because the value of the in-situ transverse compressive strength selected (Table 1) is approximately the same of the UD ply. Reconsider this analysis using a correct value for the in-situ transverse compressive strength, necessarily higher than the unidirectional transverse compressive strength. Please amend the manuscript accordingly. Line 207: the maximum stress criteria were applied to predict fibre fracture. But the original Puck fibre fracture criteria are different. It is therefore recommended to use the original criteria also in this case. Not employing them in the present work must be suitably justified.Minor issues:
Abstract, lines 23-25: it is mentioned that the model has been successfully implemented, but nothing is said about whether it has been successfully validated. Please refer to the validation results presented in this paper, and limitations of the performed analyses especially regarding the lack of an interlaminar damage model to predict delamination, which may have an important role in the selected validation test cases. Lines 76-84: The smeared crack model proposed by Camanho et al. (2013) also considers the fracture plane to induce material degradation due to transverse cracking (IFF). At least reference to this work should be included.Ref. Camanho et al. Mechanics of Materials 59 (2013) 36–49. Eq. (14): for consistency, replace T^-T by (T^T)^-1. Lines 249, 254 and 263: How the characteristic lengths are defined is not clear. Please add information about how the inter-fibre and fibre fracture characteristic lengths are determined, and whether they are different. Lines 269-270: it is mentioned that the results predicted by the original and modified Puck’s IFF criteria are almost identical for the ply in a unidirectional laminate. Please enumerate which (small) differences have been identified. They are not clear from Fig. 5. Line 303 and Table 3: to keep consistency, please use “IM7/8552”. Figure 10 and Figure 12: Not considering interlaminar damage may be the reason for the overprediction. At least some considerations about not having included interlaminar damage in the model should be included in the paper.
Overall, writing is good. Only a few corrections are recommended:
Line 38: “… failure mechanisms… due to their…”. Line 39: “… Hence, developing…”. Line 50: “… and Tsai–Wu fail”. Line 52: “Hashin proposed a set of criteria…”. Line 73: “… energy is dissipated for…”. Figure 2, caption: “… relation under simple…”. Figure 3, caption: “… relation under simple…”. Line 251: “failure mode, a quadratic…”. Line 267: “… are taken as the case study. …”. Figure 5, caption: “… relation under simple…”. Figure 7, caption: “… (a) Ply in a… (b) Ply embedded…”. Line 331: “… under uniaxial compressive loading…”. Line 351: “… in an FE software…”.Author Response
Please see the attachment.

Reviewer 3 Report
The paper proposes a continuum damage model to predict the progressive failure of a composite laminate. The model is based on a modified Puck’s failure criteria for failure initiation, uses a bi-linear law for the stiffness degradation, accounts for the so-called in situ strength of a ply in a laminate, and uses damage variables expressed in the fracture plane coordinate system. Good correlation with the experimental data is shown.
The paper is clearly organized and well written, and characterized by a satisfactory degree of novelty. Only the following minor changes are advised before publication on Materials.
Section 1 (Introduction): “IFF” is used before it is defined, which is done later at the beginning of Section 2.1. Please define IFF before its first use. Section 2.1: To prevent misunderstandings when obtaining equation (2), the last three elements of the strain vectors should be γ23, γ31, γ12 and be γnt, γlt, γnl instead of ε23, ε31, ε12 and be εnt, εlt, εnl, where εij = γij/2 Section 2.2.1: The lay-up of the “multidirectional laminate” referred to in Figure 3 and in the relevant text should be mentioned. If the lay-up does not affect the result, the reason should be stated. Section 3.1: As the previous point, the lay-up of the “multidirectional laminate” should be stated. Section 3.2: it should be stated where the mechanical properties of Tables 2 and 3 were taken from or calculated, with particular reference to the energy release rates and the parameter β. Figure 11: the remote load to which the four images refer to should be reported. In addition, it is not mentioned if there is any damage in the 45° plies (it would be expected before the laminate final failure!). Figure 13: as for Figure 11, the remote load to which the image refers to should be indicated.Author Response
Please see the attachment.

Round 2
Reviewer 2 Report
This manuscript presents a modification to a well-established set of failure criteria for unidirectional fibre-reinforced composite materials to account for thickness effects on the strength of unidirectional layers embedded in multidirectional laminates, and implementation of the proposed modified criteria in combination with a continuum damage mechanics model in a finite element context. The proposed modification is pertinent, and therefore worth publication. The authors have successfully addressed most of the issues raised by the reviewer. However, a couple of recommendations must be implemented before the manuscript is ready for publication in Materials.
Major issues:
Response to Point 1: the analysis and argumentation presented should be included in the revised manuscript (including Figure I and equations (I-V)), for the sake of completeness, for example as an appendix. Please refer to this appendix when stating that for a ply embedded in a multidirectional laminate "the predicted results are reasonable under transverse compression" (lines 177-178 and 296-297 of the revised manuscript). Response to Point 1: the authors insisted in keeping the same value in Table 1 for the in-situ transverse compressive strength. However, the in-situ transverse compressive strength cannot be lower than the transverse compressive strength of a UD ply - this is not correct. The authors are strongly advised to use, for example, the equation in Table 5 to calculate the in-situ transverse compressive strength and use this value in the analyses performed in Sections 2.2.1 and 3.1.Writing corrections and minor issues:
Line 87: “… influence of the fracture plane orientation…”. Equation (17): based on line 114, should sigma be replaced by tau? Line 291: “between the curves due to the small differences between…”. Line 319: “used in the remaining regions…”. Line 322: “cubic root of the volume of the elements. …”. Line 364: “… (see Figure 11e and f)…”. Reference 24 is not the correct reference. The correct citation should be Camanho et al. Mechanics of Materials 59 (2013) 36–49.Author Response
Please see the attachment.

Round 3
Reviewer 2 Report
This manuscript presents a modification to a well-established set of failure criteria for unidirectional fibre-reinforced composite materials to account for thickness effects on the strength of unidirectional layers embedded in multidirectional laminates, and implementation of the proposed modified criteria in combination with a continuum damage mechanics model in a finite element context. The proposed modification is pertinent, and therefore worth publication. The authors have successfully addressed the issues raised by the reviewer. Only a couple of comments need clarification before the manuscript is ready for publication in Materials.
In Figure 3, is the blue curve correct? It seems equal to the curve in Figure 2. Comparing with Figure A1, it would be expected to see the F_IFF-theta relation with a lower minimum value than in Figure 2. Please clarify and correct accordingly. In Figure A1, the ‘1.5 Y_C’ and ‘2 Y_C’ curves seem to be switched. Please clarify and correct accordingly.Author Response
Please see the attachment.
